# Effect of Sinapic Acid on Scopolamine-Induced Learning and Memory Impairment in SD Rats

**DOI:** 10.3390/brainsci13030427

**Published:** 2023-03-01

**Authors:** In-Seo Lee, Ga-Young Choi, Inturu Sreelatha, Ji-Won Yoon, Suk-Hee Youn, Sungho Maeng, Ji-Ho Park

**Affiliations:** 1Department of Gerontology (AgeTech Service Convergence Major), Graduate School of East-West Medical Science, Kyung Hee University, Yongin 17104, Republic of Korea; 2Center for Research Equipment, Korea Basic Science Institute, Cheongju 28119, Republic of Korea; 3Department of East-West Medicine, Graduate School of East-West Medical Science, Kyung Hee University, Yongin 17104, Republic of Korea

**Keywords:** sinapic acid, scopolamine, Alzheimer’s disease, long-term potentiation, learning and memory, anti-inflammatory

## Abstract

The seriousness of the diseases caused by aging have recently gained attention. Alzheimer’s disease (AD), a chronic neurodegenerative disease, accounts for 60–80% of senile dementia cases. Continuous research is being conducted on the cause of Alzheimer’s disease, and it is believed to include complex factors, such as genetic factors, the accumulation of amyloid beta plaques, a tangle of tau protein, oxidative stress, cholinergic dysfunction, neuroinflammation, and cell death. Sinapic acid is a hydroxycinnamic acid found in plant families, such as oranges, grapefruit, cranberry, mustard seeds, and rapeseeds. It exhibits various biological activities, including anti-inflammatory, anti-oxidant, anti-cancer, and anti-depressant effects. Sinapic acid is an acetylcholine esterase inhibitor that can be applied to the treatment of dementia caused by Alzheimer’s disease and Parkinson’s disease. However, electrophysiological studies on the effects of sinapic acid on memory and learning must still be conducted. Therefore, it was confirmed that sinapic acid was effective in long-term potentiation (LTP) using organotypic hippocampal segment tissue. In addition, the effect on scopolamine-induced learning and memory impairment was measured by oral administration of sinapic acid 10 mg/kg/day for 14 days, and behavioral experiments related to short-term and long-term spatial memory and avoidance memory were conducted. Sinapic acid increased the activity of the field excitatory postsynaptic potential (fEPSP) in a dose-dependent manner after TBS, and restored fEPSP activity in the CA1 region suppressed by scopolamine. The scopolamine-induced learning and memory impairment group showed lower results than the control group in the Y-maze, Passive avoidance (PA), and Morris water maze (MWM) experiments. Sinapic acid improved avoidance memory, short and long-term spatial recognition learning, and memory. In addition, sinapic acid weakened the inhibition of the brain-derived neurotrophic factor (BDNF), tropomyosin receptor kinase B (TrkB) and the activation of prostaglandin-endoperoxide synthase 2 (COX-2) and interleukin 1 beta (IL-1β) induced by scopolamine in the hippocampus. These results show that sinapic acid is effective in restoring LTP and cognitive impairment induced by the cholinergic receptor blockade. Moreover, it showed the effect of alleviating the reduction in scopolamine-induced BDNF and TrkB, and alleviated neuroinflammatory effects by inhibiting the increase in COX-2 and IL-1β. Therefore, we showed that sinapic acid has potential as a treatment for neurodegenerative cognitive impairment.

## 1. Introduction

In modern times, neurodegenerative diseases have become a serious global health problem. Alzheimer’s disease (AD), first reported by Germany’s Alois Alzheimer in 1906, is the most common neurodegenerative disease, characterized by initial memory impairment and cognitive decline, and it affects behavior [1]. Therefore, AD, an acquired cognitive impairment that progresses to the extent that it affects daily life activities, has a high mortality rate in older adults [2]. The number of patients with AD is increasing worldwide and is expected to more than triple by 2050 as the population ages further [3,4,5]. In contrast, no clear treatment or method has been found for AD, and recent studies have shown that several mechanisms are involved in the deposition of β- amyloid (Aβ) plaques, Aβ aggregation, neuronal death, cholinergic nervous system diseases, the formation of NFTs with abnormal phosphorylated tau, oxidative stress, neuroinflammation, and genetic factors [6,7]. Therefore, anti-oxidants and anti-inflammatory drugs are effective adjuvant therapies for AD [8,9].

Choline neurons in the hippocampus and medial septum (MS) are involved in learning and memory action [10]. Therefore, scopolamine (SCOP), a muscarinic acetylcholine (Ach) receptor antagonist, impairs short-term and spatial memory in animal models and humans in a manner similar to AD pathology [11,12]. In addition, SCOP has been widely used to induce learning and memory disorders in animal models by increasing Acetylcholinesterase (AChE) levels in the hippocampus and reducing brain-derived neurotrophic factor (BDNF) [13]. Furthermore, SCOP treatment improved the neuroinflammatory conditions caused by the release of interleukin 1 beta (IL-1β) and tumor necrosis factor α (TNF-α), and increased the expression of induced nitric oxide synthase and mRNA [12,14]. Moreover, neuronal cell death is induced along with a cholinergic deficit, showing neuronal cell death in neurodegenerative diseases [9].

Polyphenols are phenolic compounds found in plants and are believed to be involved in the defense against harmful oxidative damage. Among the polyphenol compounds, hydroxycinnamic acid and its derivatives are well-known chemical groups that have important biological functions such as anti-inflammatory and anti-oxidant activities [15]. The beneficial effects of these compounds as preventive or therapeutic agents have already been demonstrated in various diseases, such as inflammatory damage and cancer [16,17]. Among the hydroxycinnamic acids, sinapic acid (SA; 2E-3-4-Hydroxy-3,5-dimethoxyphenylprop-2-enoic acid; Figure 1), a natural herb phenolic acid compound, is present in herbs, such as oranges, grapefruits, cranberries, mustard seeds and rapeseeds [18]. SA has been reported to prevent SCOP-induced cognitive impairment in the memory of a rat model [19]. It showed anti-inflammatory activity in carbon tetrachloride-induced acute liver damage and significantly reduced proinflammatory cytokines levels, such as IL-1β and TNF-α [20,21]. Therefore, SA relieves and prevents oxidative stress, neuroinflammation, and cholinergic deficits [22].

Long-term potentiation (LTP) is defined as the ability to convert electrical stimuli into chemical signals, thereby activating the pre-synaptic and post-synaptic mechanisms, resulting in a continuous increase in synapses [23]. LTP is induced by the activation of N-methyl-D-aspartate (NMDA) type glutamate receptors via the simultaneous activity of the pre- and post-synaptic neurons [24]. LTP has been proposed as a synaptic mechanism of cognitive function because NMDA receptor-dependent LTP can constitute the cellular substrate of learning and memory [25,26].

In this study, the neuroprotective and anti-inflammatory effects of SA in SCOP-induced learning and memory impairment models were evaluated through electrophysiological, behavioral, and immunological experiments.

## 2. Materials and Methods

### 2.1. Materials

Sinapic acid (SA, D7927), scopolamine hydrobromide (SCOP, S0929), 4-(2-hydroxyethyl)-1-piperazineethanesulfonic acid (HEPES, H4034), L-glutamine (G-8540), D-glucose (G7021), sodium chloride (NaCl, S7653), potassium chloride (KCl, P5405), calcium chloride dihydrate (CaCl_2_, C7902), magnesium chloride hexahydrate (MgCl_2_, M2393), sodium carbonate (NaHCO_3_, S5761) and penicillin-streptomycin (P4333) were purchased from Sigma-Aldrich (St. Louis, MO, USA). Sodium dihydrogen phosphate (NaH_2_PO_4_, AB15ES) was obtained from DAEJUNG (Siheung, Republic of Korea). Minimum essential medium (MEM, LM 007-01) and Hank’s balanced salt solution (HBSS, LB 003-01) were purchased from Welgene (Kyungsan, Republic of Korea). Donor horse serum (HS, S0910-500) was obtained from Biowest (Nuaillé, France). All of the reagents used in the experiments were research grade products.

### 2.2. Experimental Animals

Seven day old Sprague-Dawley (SD) rats were used in an organotypic hippocampal slice culture (OHSCs), and 5 week old male SD rats (weighing 140 ± 10 g) were used in the behavioral test. The rats used in the experiment were purchased from Saerone Bio, Inc (Uiwang, Republic of Korea). A standard diet for feeding during the entire experimental period was provided by Saeron Bio Inc. The indoor conditions were maintained at 12 h intervals in a light/dark cycle (lights on at 8 a.m., lights off at 20 p.m.) and a temperature of 25 ± 1 °C with 50 ± 5% humidity. The cage contained two rats, each with sufficient space, and supplied with tap water and a standard diet during the experiments. All of the animal experiments were approved by the Institutional Care and Use Committee (KHGASP-22-052) of Kyung-Hee University and were performed according to the guidelines of the Council of the National Institutes of Health Guide for the Care and Use of Laboratory Animals. Efforts were made to minimize the experience of pain in the animals during the experiment.

### 2.3. Electrophysiological Experiments

#### 2.3.1. Organotypic Hippocampal Slice Cultures (OHSCs)

The brains were quickly extracted from the seven day old SD rats after decapitation and immediately immersed in a cold HBS medium with 20 mM HEPES. The hippocampus extracted from the temporal lobe of the brain was cut to a thickness of 350 µm using a tissue chopper (Mickle Laboratory Engineering Co., Gomshall, UK). Each of the six well plates filled with 1 mL of culture medium (MEM 50% *v*/*v* + HBSS 25% *v*/*v* + HS 25% *v*/*v*, supplemented with 20 mM HEPES, 5.25 g/L D-glucose, 1 mM L-glutamine, and 1% *v*/*v* penicillin-streptomycin; pH = 7.1) was added with 0.4 µm membrane culture inserts (Mi8llicell-CM); and 5–6 tissue slices were placed on each insert. The culture medium was changed every two days, and the slices were incubated at 35 °C with 5% CO_2_ for 12–14 days before the experiment.

#### 2.3.2. Preparation of Organotypic Hippocampal Tissue on the Micro-Electrode Array (MEA) Probes

A single stabilized hippocampal slice tissue was carefully removed from a membrane of the insert with a soft brush and dropped into artificial cerebrospinal fluid (aCSF; 114 mM NaCl, 3 mM KCl, 1.1 mM NaH_2_PO_4_, 2.5 mM NaHCO_3_, 25 mM D-glucose, 2 mM CaCl_2_, 1.3 mM MgCl_2_, and 20 mM HEPES; pH 7.4). The isolated hippocampal slice tissue was attached to an 8 × 8 MEA with 10 µm-diameter electrodes, which was coated with 0.01% polyethylenimine (PEI). The MEA system consists of a stimulator, amplifier, temperature control unit, and computer software for data acquisition [27]. The slice tissue was stabilized in aCSF for 30 min at 33 °C with 95% O_2_ and 5% CO_2_ gas aeration.

#### 2.3.3. Induction of Long-Term Potentiation (LTP) and Treatment for Hippocampal Slice

Bipolar electrical stimulation was applied to the stratum radiatum region of the cornu ammonis (CA) 2 and CA3 to induce the Schaffer collateral and commissural pathways. The intensity of the bipolar stimulation was set to 160 μA, 240 ms per phase, and optimized to provide 40–65% of the maximum tissue response. The theta burst stimulation (TBS) consisted of 300 biphasic pulses and three trains at 100 Hz, for 1 sec each, at 5 min intervals. Each experiment to induce LTP was planned with a total 90 min protocol and consisted of 30 min of field excitatory postsynaptic potentials (fEPSPs) recordings, 10 min of TBS, and 50 min of fEPSP measurements after TBS. The Schaffer side and commissural pathways were selected by the morphological structure of the hippocampal tissue and the appropriate reaction to bipolar electrical stimulation. During the experiment, fresh aCSF was continuously injected into the hippocampal slices at 3 mL/min. In addition, the SA and SCOP were treated with aCSF 10 min after the start of recording. 

#### 2.3.4. Electrophysiology Data Processing

All of the data were sampled from 60 recording channels at 25 kHz using the Recorder-Rack software. The experimental data consisting of analog MEA signals with MC_Rack (v.3.2.1.0, Multi-Channel Systems, Reutlingen, Germany) were converted to digital form, of which EPSPs over 80 mV were selected using the trigger mode. A custom MATLAB (v.7.0.1, Mathworks, Inc., Natick, MA, USA) program was used to remove the stimulus artifacts and integrate the evoked field potential trajectory, as reported previously. 

### 2.4. Behavioral Tests

#### 2.4.1. Experimental Design

The SA and SCOP were prepared by dissolving in deuterium-depleted water (DDW). The SA was administered at a dose of 10 mg/kg/day per oral (p.o.) administration. The SCOP was administered by intraperitoneal (i.p.) injection at a dose of 1.5 mg/kg/day. Each group (Control, SCOP, SA and SCOP + SA) consisted of six rats selected randomly. The SA was pretreated for 10 days before the start of the behavioral test. After preprocessing, the SA and SCOP were administered to each rat, 60 and 30 min prior to the start of individual tests during the behavioral experiments. Adaptation, pretreatment, and treatment with behavioral tests, including Y-maze, Passive avoidance (PA), and Morris water maze (MWM) test were performed for 21 days, and all of the rats were sacrificed on the last day.

#### 2.4.2. Y-Maze Test

The Y-maze test was performed to measure short-term spatial memory. The Y-shaped maze had a structure in which three-way arms (45 cm × 35 cm × 10 cm) intersected at 120°. A rat was placed at the center of the arm’s connection point and was allowed to operate freely in the Y-maze for 10 min without any derivatives or obstacles. All four of the rats’ paws had to be inside each arm for approval, and the spontaneous alternation was measured, which was a consecutive entry into the three arms. The following formula was used to calculate the spontaneous alternation percentage [28]:  [(number of spontaneous alternations)/(total number of arm entries − 2)] × 100

#### 2.4.3. Passive Avoidance (PA) Test

A PA test was performed to measure the memory of long-term avoidance by fear. The shuttle box, a device consisting of two chambers of the same size (17 cm × 12 cm × 15 cm), was installed on one side and was separated by a guillotine door (9 cm × 17 cm). Several stainless-steel grids were installed on the floor and connected to an electric shock stimulator, which transmitted the electric shock when a rat enters a dark area. The experiment was conducted with two different trials: a training session for fear acquisition and a retention session to check if the memory of the fear remains. In the training session, the rats were initially placed in the light area and allowed to adapt for 60 sec after the door between the two areas was opened. When all four of the rats’ paws crossed the dark area, the door closed, and an electric shock was transmitted. After 20 sec, the rat was removed from the dark area and returned to its cage. The fear memory retention session was measured 24 and 72 h after the training session. The rats were placed in the light area for 60 sec, and the door opened. After the step-through latency time entering the dark area was measured for 600 sec.

#### 2.4.4. Morris Water Maze (MWM) Test

The Morris water maze (MWM) test was used to evaluate long-term spatial learning and memory. A circular water tank of 180 cm × 45 cm (diameter × height) was filled with 25 ± 1 °C of water and made opaque by adding non-toxic white paints. An escape platform was randomly placed on one of the four quadrants, divided into the same size, and submerged 1 cm below the water surface to hide. Four visual cues of different shapes were randomly designated and placed in the north, south, east, and west. The rats were trained in four different areas, every day for four days, during the training period. The latency required to reach the submerged platform was recorded. When the rats were able to reach on the platform within 60 sec, they were allowed to rest on the platform for 20 sec. If the rats did not reach the platform, they were guided to the platform by an experimenter for 20 sec. On the test session, the rats were placed in a random area other than the training session on day 4 and were allowed to navigate for 90 sec without the platform. All of the experiments were recorded using a video camera (SHC-650A; Samsung, Suwon, Republic of Korea) installed on the top of the tank. Furthermore, the results were analyzed using the SMART video tracking software.

### 2.5. Western Blot Analysis

First, tissues from rats in the Control, SCOP, SA and SCOP + SA groups were analyzed using western blotting. After the behavioral experiments, the brain was dissected, and the hippocampus was quickly extracted and stored in a −80 °C deep freezer. Next, the extracted hippocampus was homogenized by sonication in RIPA buffer containing phosphatase inhibitors and a complete protease inhibitor cocktail. Then, the hippocampus extracts were stored on ice for 30 min and then centrifuged at 20,817× *g* for 15 min at 4 °C. The protein concentration of the supernatant was measured using a Bradford protein assay, and equal amounts of protein were separated on 10% SDS-PAGE gel and transferred to PVDF membranes. Subsequently, the membranes were blocked in TBS with 0.1% Tween 20 containing 5% dry skim milk for 1 h and incubated overnight at 4 °C in 5% skim milk containing primary anti-bodies. The primary anti-bodies used were rabbit monoclonal for brain-derived neurotrophic factor (BDNF; ab108319, Abcam), rabbit polyclonal for tropomyosin receptor kinase B (TrkB; ab18987, Abcam), rabbit monoclonal for prostaglandin-endoperoxide synthase 2 (COX-2; ab179800, Abcam) and rabbit monoclonal for interleukin 1 beta (IL-1β; ab234437, Abcam). Next, membranes were washed three times in TBST for 15 min, and incubated in 5% skim milk containing secondary anti-bodies for 1 h at 25 °C. After the final washing, the bands were visualized with an EzWestLumi plus ECL solution (WSE-7120S, ATTO Co., Tokyo, Japan) and an image detection system. All of the experiments were repeated at least three times using different tissue sample batches, and the results were reproducible.

### 2.6. Statistical Analysis

All of the data are expressed as mean ± standard error of the mean (SEM). Statistical analysis was performed using the SPSS software (version 26.0; IBM SPSS Statistics Inc., Chicago, IL, USA). Significant differences in the average values were analyzed using a one-way ANOVA with the least significant difference (LSD) test (*p* < 0.05). 

## 3. Results

### 3.1. SA Strengthened Hippocampal CA1 fEPSP

SA (1, 10, and 100 μM) was applied to the hippocampal CA1 region for LTP, and the CA1 region was measured. The time-dependent percentage change in the fEPSP activity and mean percentage of the fEPSP from 30 to 40 min after TBS were analyzed. SA increased the total fEPSP activity after TBS (Figure 2A). The average post-TBS fEPSP activity was significantly different between the treatment groups at 30 to 40 min [F(3, 12) = 547.792, *p* < 0.001; Figure 2B]. In all of the dose tests, SA increased the average fEPSP activity in a dose-dependent manner compared to that in the control group (1 μM; *p* < 0.001, 10 μM; *p* < 0.001, 100 μM; *p* < 0.001).

### 3.2. SA Restored Suppressed LTP by SCOP

Next, the effect of SA on LTP impairment induced by SCOP, which is considered a representative cause of memory impairment and neuroinflammation, was tested. The tissues were treated with 300 μM SCOP and 10 μM SA + 300 μM SCOP. The total fEPSP activity in the SA + SCOP group showed significantly higher activity than the SCOP group (Figure 3A). The average value of the fEPSP activity between 30 to 40 min after TBS also showed differences [F(3, 12) = 1337.981, *p* < 0.001, Figure 3B]. The SA + SCOP co-treatment group increased (*p* = 0.001), and the SCOP group decreased (*p* < 0.001), compared with the control group. According to these results, SA increased the LTP of the hippocampal CA1 in a dose-dependent manner and improved the LTP suppressed by SCOP (Figure 2 and Figure 3).

### 3.3. SA protected Short-Term Spatial Memory in SCOP-Induced Rats

The Y-maze test was performed to confirm the effects of SCOP and SA on short-term spatial memory. The percentage of spontaneous alternation was significantly different among the treatment groups [F(3, 12) = 11.738, *p* = 0.001; Figure 4]. The SCOP group showed a decrease in spontaneous alternation compared with the control group (*p* < 0.01), and the SCOP + SA group showed an increase in alternation behavior compared with the SCOP group (*p* < 0.01).

### 3.4. SA Protected Long-Term Avoidance Memory in SCOP-Induced Rats

A passive avoidance test was performed to confirm the effects of SCOP and SA on long-term avoidance memory (Figure 5). The step-through latency was measured at 24 h, and 72 h later, the fear memory was formed by electric foot shock. In the acquisition trial, there was no significant difference in the step-through latency between the groups [F(3, 12) = 0.697, *p* = 0.571], and was the same at 24 h [F(3, 12) = 0.446, *p* = 0.725]. However, in the retention test conducted at 72 h, the difference in the step-through latency was measured compared to the SCOP group [F(3, 12) = 2.605, *p* = 0.100]. In the retention test at 72 h, the SCOP group showed a decrease compared to the control group (*p* = 0.058). However, the SA + SCOP group showed increased avoidance memory compared with the SCOP group (*p* < 0.05).

### 3.5. SA Protected Long-Term Spatial Memory in SCOP-Induced Rats

The effects of SCOP and SA on long-term spatial learning and memory were measured using the Morris water maze (MWM) test. The MWM test was performed to confirm the effect of SCOP and SA on long-term spatial memory. In all of the experimental groups, the latency to platform (escape latency) decreased during training sessions of four days [F(3, 12) = 7.704, *p* = 0.004; Figure 6A]. As a result, the control group decreased from 58.30 ± 0.98 sec to 24.87 ± 2.05 sec and the SA group decreased from 54.74 ± 3.05 sec to 21.32 ± 1.55 sec. However, the SCOP group showed a weak decline compared to other groups, from 55.87 ± 2.40 sec to 48.94 ± 4.41 sec (*p* < 0.01). In comparison, the SCOP + SA group decreased from 58.13 ± 0.97 sec to 30.69 ± 6.47 sec, and showed the same level of long-term spatial learning and memory as the control group, compared to the SCOP group (*p* < 0.05). In the probe trial, the target quadrant time between each group was significantly different [F(3, 12) = 4.077, *p* = 0.033; Figure 6B]. The SCOP group showed a reduction compared with the control group (*p* = 0.051). In the SCOP + SA group, there was an increase in the target quadrant time compared to that in the SCOP group (*p* < 0.05).

### 3.6. SA Increased Neurotrophic Activity in SCOP-Induced Rats

Density measurement analysis of the western blot showed a significant difference in the expression of BDNF [F(3, 12) = 9.289, *p* = 0.002, Figure 7A] and TrkB [F(3, 12) = 655.902, *p* < 0.001, Figure 7B] in the hippocampus of the experimental groups. In the SCOP group, the expression of BDNF decreased compared to the control group (*p* = 0.089). However, in the SCOP + SA group, the expression of BDNF was higher than the SCOP group (*p* = 0.518). The expression of TrkB was also lower in the SCOP group than in the control group (*p* < 0.001) and, in contrast, the SCOP + SA group increased significantly compared to the SCOP group (*p* = 0.001).

### 3.7. SA Relieved Neuroinflammation in SCOP-Induced Rats

The difference in COX-2 expression [F(3, 12) = 8.239, *p* = 0.003, Figure 8A] and IL-1β [F(3, 12) = 2.711, *p* = 0.092, Figure 8B] was used to measure the inflammatory relief effect of SA by western blotting. In the SCOP group, the expression levels of COX-2 (*p* < 0.01) and IL-1β (*p* < 0.05) were significantly increased compared with the other groups that were intended to be effective in the inflammation of the hippocampus. In contrast, the SCOP + SA group showed reduced protein levels of COX-2 (*p* < 0.01) and IL-1β (*p* = 0.186) compared to those in the SCOP group.

## 4. Discussion

This study investigated whether SA alleviates SCOP-induced learning and memory impairment by performing electrophysiological and behavioral experiments. As a result, SA reduced the SCOP-induced disturbances in the LTP, short and long-term spatial memory impairment, and avoidance memory impairment. In addition, the expressions of BDNF and TrkB improved, and the activities of IL-1β and COX-2 were suppressed.

LTP has mainly been used as an experimental model to measure mechanisms related to learning and memory [23,29]. In addition, LTP has been used to study neurodegenerative diseases, such as AD. Cholinergic blockade by anti-cholinergic SCOP induces a decrease in LTP [26]. This study also showed that hippocampal slice tissue treated with SCOP showed a significant reduction in post-TBS fEPSP compared to the control group. In previous studies, behavioral experiments showed that SA was effective in alleviating learning and memory dysfunction. However, the effect of SA on LTP has not been verified. This study showed that SA treatment increased LTP dose-dependently at concentrations of 1, 10, and 100 μM compared to the control group. However, for capacities of 10 and 100 μM, the increase became similar over time after TBS. This means that the concentration of 10 μM may be the limitation point of effect. Therefore, we evaluated the protective effect of SA on SCOP-induced LTP inhibition using 10 μM of SA and confirmed that treatment with SCOP + SA reduced LTP damage. 

This study selected the method of treating SCOP and SA together. As another method, the recovery and prevention effects of SA can be proven by administering SCOP first, followed by SA, or by treating with SA first and then SCOP. However, in the LTP experiments, the effective fEPSP activity is measured 30 min after high-frequency stimulation. Even if SCOP and SA are treated separately, they are eventually processed before high-frequency stimulation. Therefore, it is not clear whether the result is different from processing SCOP and SA simultaneously. Due to these limitations on the protocol of LTP, many studies have applied the simultaneous processing method when using the organotypic hippocampal slice culture [30,31]. In addition to experiments of injecting drugs into aCSF while recording, methods of incubation with SCOP treatment, measurement of LTP, incubation with SA treatment, and measurement of LTP can also be attempted [32]. Therefore, we would design an experiment on the effect of SA by dividing sequences into SCOP first/SA post and SA first/SCOP post and processing in 30, 45, and 60 min time units in the follow-up study.

The Y-maze test was performed to measure short-term spatial memory, and the Morris water maze (MWM) test was performed to measure long-term spatial learning and memory [33,34]. In a previous study, SA significantly increased the escape latency of a cognitive impairment model induced by intracerebroventricular streptozotocin (ICV-STZ) in the executed probe session of the MWM test conducted after 21 days of administration, compared to the control group. However, in the case of the target quadrant time, the results differed depending on capacity [20]. In our results, SA not only increased short-term memory in the Y-maze test but also improved long-term learning and memory. 

The PA test was performed to measure avoidance memory [35]. Previous studies have measured the latency by administering SA on amyloid beta 1–42 protein-induce cognitive impairment mice. This showed a dose-dependent difference, and only the groups treated with 10 mg/kg of 1, 3, and 10 mg/kg doses showed a significant difference for the disorder group [21]. This is similar to the results of the PA test in the cognitive impairment model mice induced by kainic acid. Of the doses of 1, 3, and 10 mg/kg, only groups of 3 and 10 mg/kg showed significant latency, and 10 mg/kg showed greater difference [36]. These results indicate that SA alleviates concentration-dependent hippocampal disorders associated with learning and memory. In our study, memory and learning disorders were induced through SCOP processing, and short and long-term spatial memory were maintained by SA processing. In the case of avoidance memory, no significant difference was found at 24 h. However, a significant difference was found at 72 h. Therefore, SA treatment improves long-term avoidance memory. 

Nuclear factor erythroid 2-related factor 2 (NRF2) and nuclear factor-KB (NF-KB) are the two main transcription factors that regulate the cellular response to oxidative stress and inflammation, respectively. The absence of NRF2 can exacerbate NF-KB activity and increase cytokine production, whereas NF-KB can regulate NRF2 transcription and activity, affecting target gene expression [37]. The extracellular region of TrkB in vivo is known to bind BDNF with a high affinity [38]. BDNF is a well-known indicator of synaptic function regulation, neural survival, and differentiation, and can have therapeutic implications in various neurodegenerative diseases [39,40]. Therefore, these proteins also play pivotal roles in improving hippocampal LTP and synaptic plasticity. In previous studies, the cell survival rate, mribonucleic acid (mRNA) expression of BDNF, and relative expression of TrkB protein were significantly increased by treatment with SA [41]. In this study, the expressions of BDNF and TrkB in the hippocampus were reduced by SCOP. In contrast, SA increased the expression of the two proteins, particularly in the case of TrkB. There was a significant difference when co-treatment with SCOP and SA was performed. Therefore, SA can improve the growth of neuro-growth factor-induced neurons and neuroplasticity through NRF2 activation [42].

Infectious cytokines such as IL-1β are potent NF-KB activators. COX-2 is induced by infectious cytokines and stress factors. COX-2 is an important target for anti-inflammatory drug development, and inhibition of COX-2 is thought to mediate the therapeutic action of nonsteroidal anti-inflammatory drugs (NSAID) [43]. In previous studies, SA significantly reduced oxidative stress and exhibited anti-inflammatory effects [44,45]. In this study, it was confirmed that inflammation was suppressed by IL-1β and COX-2, especially COX-2, and there was a significant difference when SA was treated with SCOP. This suggests that SA suppresses oxidative stress and inflammation by inhibiting NF-kB signaling.

Polyphenols are bioactive compounds prevalent in plants and have important biological functions, such as anti-oxidant activity and anti-inflammatory effects [46,47,48]. SA has been proposed as a strong anti-oxidant, and its effect is described to be higher than that of ferulic acid and sometimes similar to that of caffeic acid [49,50]. The relevance of SA to cell protection and oxidation-related diseases has already been reported because of its peroxynitrite scavenging activity [51,52]. According to previous studies, SA alleviated avoidance memory impairment in Kainic acid-induced AD model mice [36]. In addition, in the ICV-STZ-induced cognitive impairment model, ACh levels were restored through the upregulation of ChAT protein expression to alleviate cognitive impairment [20]. Although SA has been proven to be an effective anti-oxidant and anti-inflammatory agent, few studies related to the effect of SA on memory and learning have been reported. Moreover, AD is a multidisciplinary disease, and several factors were involved, and among them, active discussions were the factors of oxidative stress and neuroinflammatory factors. Therefore, the results of this study show that SA mitigates neuroinflammation-induced learning and memory impairment in AD.

## 5. Conclusions

Our results suggest that SA attenuates the SCOP-induced deterioration of short and long-term spatial memory and avoidance memory in rats. These effects were associated with the hippocampal induction of BDNF and TrkB expression and the enhancement of hippocampal synaptic activity. In addition, SA was affected by inhibiting inflammation by reducing COX-2 and IL-1β. Therefore, SA may be a possible agent for preventing and treating AD or learning and memory-deficit disorders.

## Figures and Tables

**Figure 1 brainsci-13-00427-f001:**
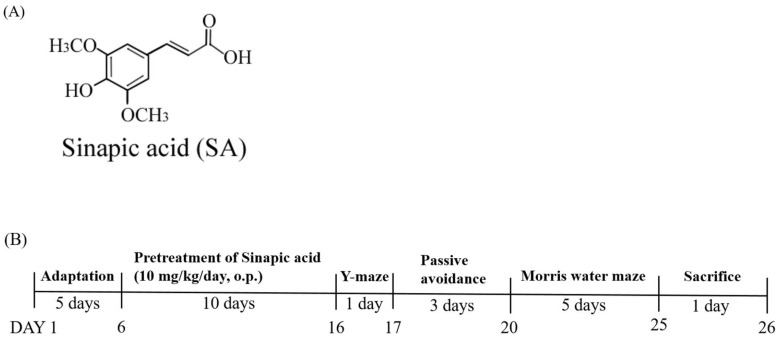
Experimental design. (**A**) Two-dimensional chemical structure of sinapic acid (SA). (**B**) Time schedule of the behavioral test.

**Figure 2 brainsci-13-00427-f002:**
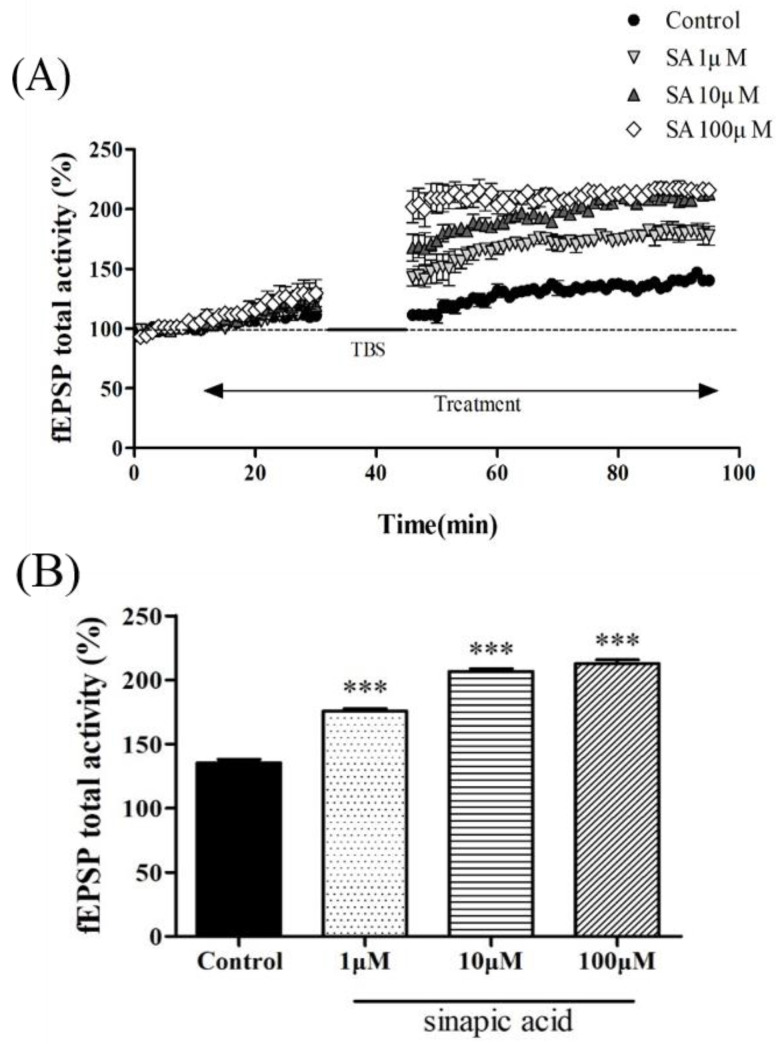
The effect of SA on long-term potentiation of the hippocampus. (**A**) fEPSP total activity (%) induced by TBS while treatment of SA in the organotypic cultured hippocampus. (**B**) Average of fEPSP total activity of organotypic cultured hippocampus treated with SA (1, 10, and 100 μM) between 30 to 40 min after TBS. n = 4. *** *p* < 0.001 versus the control group. Analyzed by one-way ANOVA with LSD test.

**Figure 3 brainsci-13-00427-f003:**
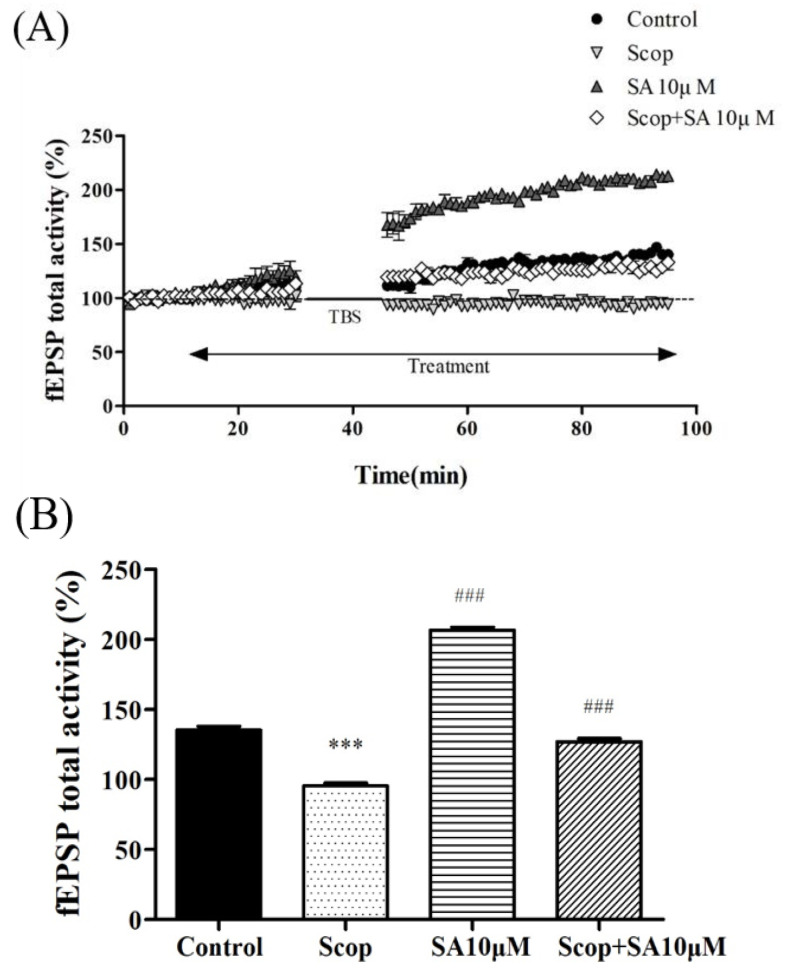
The effect of SA on long-term potentiation in SCOP-treated hippocampus. (**A**) fEPSP total activity (%) change induced by TBS while treatment of 10 μM of SA and 300 μM of SCOP in the organotypic cultured hippocampus. (**B**) Average of fEPSP total activity of 30 to 40 min after TBS during treatment of 10 μM of SA + 300 μM of SCOP in organotypic cultured hippocampus. n = 4. *** *p* < 0.001 vs. the control group. ^###^
*p* < 0.001 vs. the SCOP group. Analyzed by one-way ANOVA with LSD test.

**Figure 4 brainsci-13-00427-f004:**
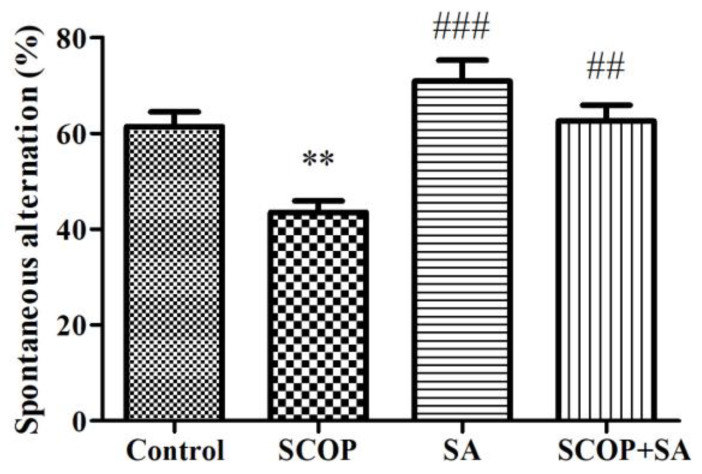
The effect of SA on short-term spatial memory in SCOP-treated. fEPSP total activity (%) change induced by TBS while treatment with 10 μM SA and 300 μM SCOP in the organotypic cultured hippocampus. n = 4. ** *p* < 0.01 vs. the control group. ^###^
*p* < 0.001, ^##^
*p* < 0.01 vs. the SCOP group. Data were analyzed by one-way ANOVA with LSD test.

**Figure 5 brainsci-13-00427-f005:**
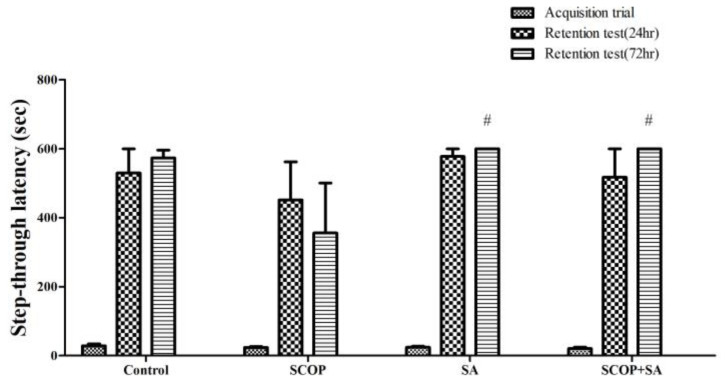
The effect of SA on long-term avoidance memory in SCOP-treated. Step-through latency was measured in the acquisition trial, retention at 24 h and retention at 72 h in the PA test. During the acquisition trial, an electric foot shock was delivered. n = 4. ^#^
*p* < 0.05 vs. the SCOP group at 72 h. Data were analyzed by one-way ANOVA with LSD test.

**Figure 6 brainsci-13-00427-f006:**
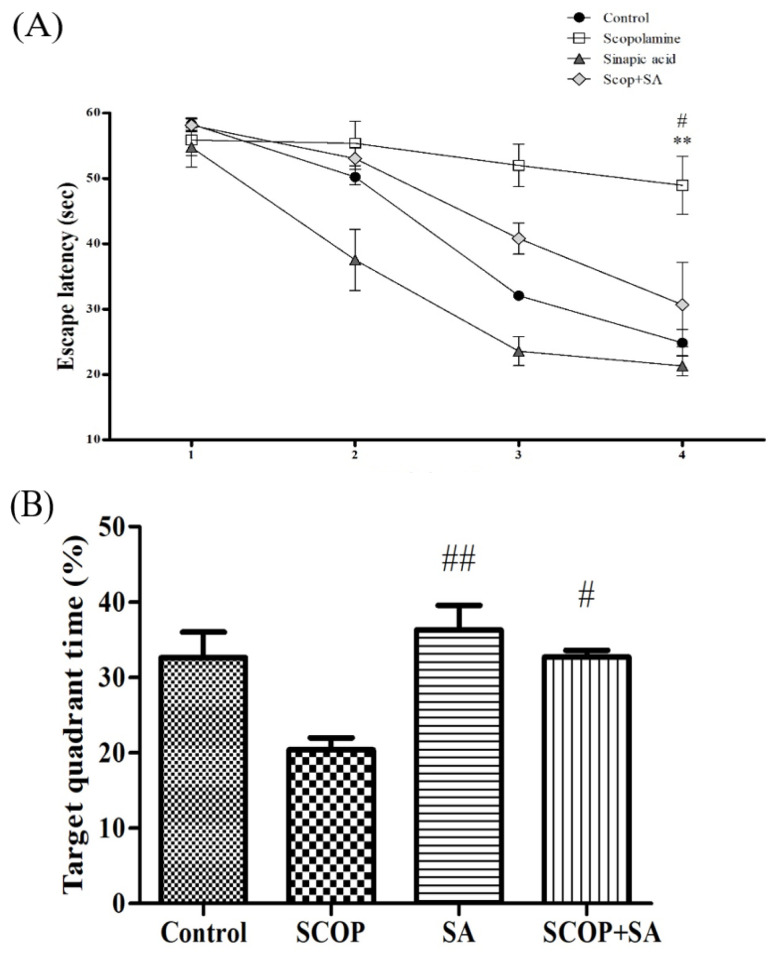
The effect of SA on long-term spatial memory in SCOP-treated. (**A**) Escape latency in the MWM test during 4 days of the training session. (**B**) Percent of time in the target zone during 60 s of the probe trial. n = 4. ** *p* < 0.01 vs. the control group. ^##^
*p* < 0.01, ^#^
*p* < 0.05 vs. the SCOP group. Analyzed by one-way ANOVA with LSD test.

**Figure 7 brainsci-13-00427-f007:**
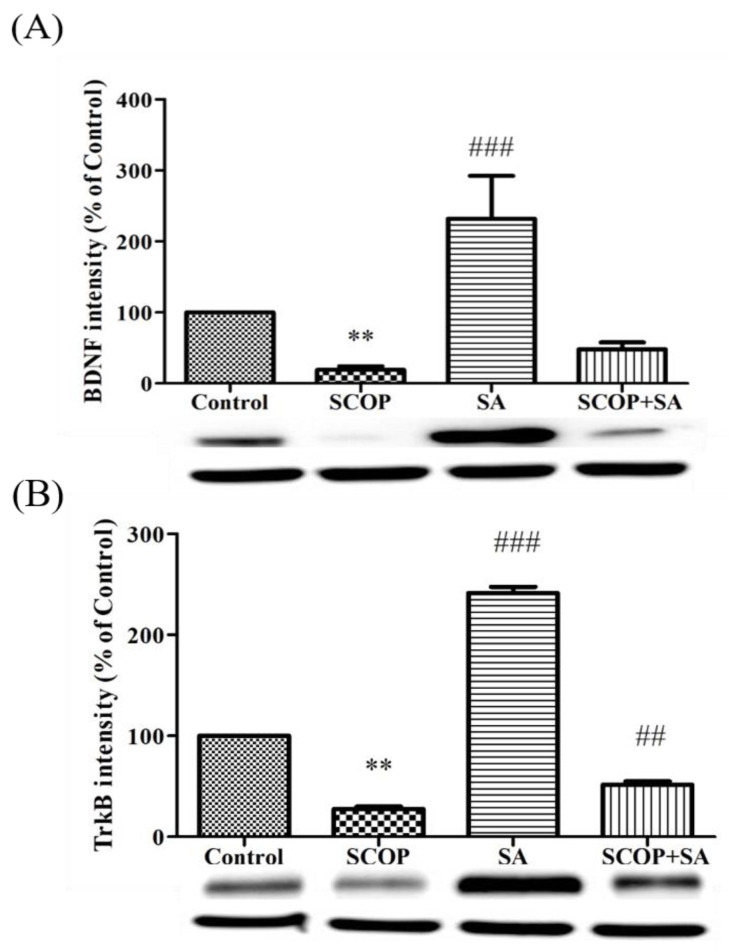
The effect of SA on the protein expression of BDNF and TrkB in SCOP-induced rat by western blot. (**A**) Representative image of immunoblotting and band quantification of BDNF/GAPDH. (**B**) Representative image of immunoblotting and band quantification of TrkB/GAPDH. All data showed as mean ± SEM. n = 4. ** *p* < 0.01 vs. the control group. ^###^
*p* < 0.001, ^##^
*p* < 0.01 vs. the scopolamine group. Analyzed by one-way ANOVA with LSD test.

**Figure 8 brainsci-13-00427-f008:**
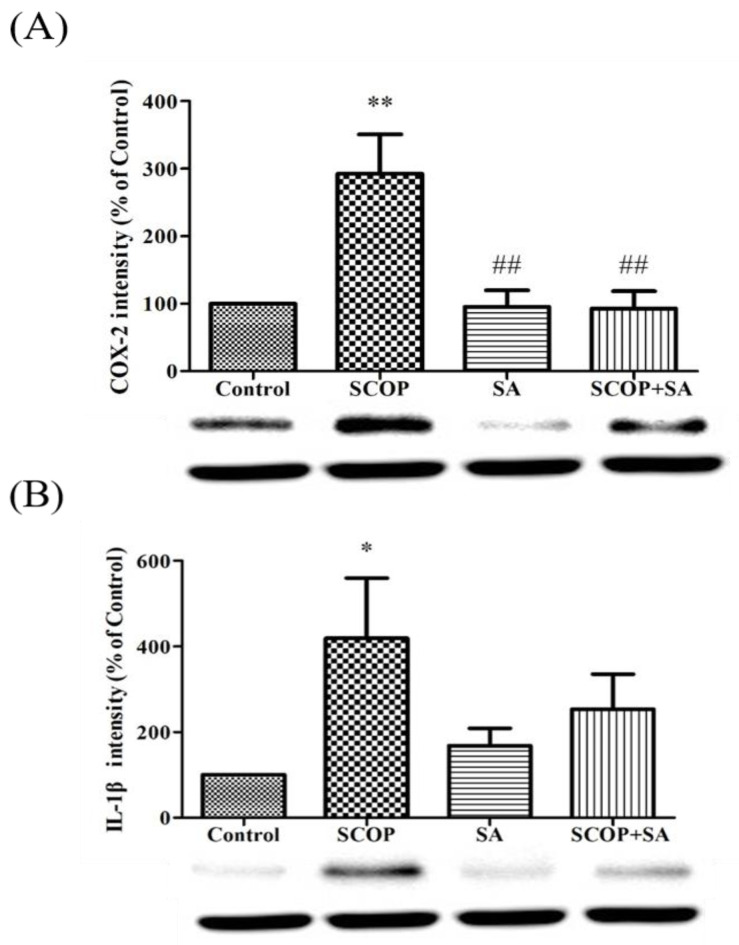
The effect of SA on the protein expression of COX-2 and IL-1β in SCOP-induced rat by western blot. (**A**) Representative image of immunoblotting and band quantification of COX-2/GAPDH. (**B**) Representative image of immunoblotting and band quantification of IL-1β/GAPDH. All data showed as mean ± SEM. n = 4. * *p* < 0.05, ** *p* < 0.01 vs. the control group. ^##^
*p* < 0.01 vs. the SCOP group. Analyzed by one-way ANOVA with LSD test.

## Data Availability

The data used to support the findings of this study are available from the corresponding author upon request.

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
