# Peer review of "Effect of Sinapic Acid on Scopolamine-Induced Learning and Memory Impairment in SD Rats"

_brainsci, 2023, doi:10.3390/brainsci13030427_

Round 1

Reviewer 1 Report

The English of the article has to be corrected in the whole article. It is difficult to sometimes understand what is meant by the text. 

The article is a study on the effect of a compound Synapic from o.a grape juice on the scopolamine induced memory loss.

Both long and short term memory is tested as well as spatial memory. 

The data show that Synaptic Acid (SA) can partially prevent the effect of Scopolamine induced memory loss. This is an interesting study that could have benefit for a number of CNS diseases. 

The questions that remain open to me with the study is the following.

In the studies, Synaptic acid is given together with Scopolamine. Grape Juice and SA has been shown to have an effect on CYP enzymes. Maybe the effect on memory os not due to inflammation or other effects but because of the inhibition of CYP enzymes. Scopolamine is interacting with CYP3A4 one of the targets of SA. The authors should test if the effect of SA is not due to the effect on the degradation or activation of Scopolamine.

One other test that should be added is to test if SA has the same effect when not given at the same time, but where first scopolamine is inducing an effect on memory and than SA is given. to show that the effect of SA is on the biological processes like inflammation and COX. 

Author Response

Response to Reviewer 1 Comments

We thank you for your time and consideration of our manuscript. Below we address your comments and list of changes we made to our manuscript according to your reports. The original comments are provided in blue color, whereas our answers are given in black. The appropriate changes made in the revised manuscript are highlighted.

We have taken the opportunity to make some other changes in order to make corrections and refine the quality of the manuscript.

Point 1: The English of the article has to be corrected in the whole article. It is difficult to sometimes understand what is meant by the text.

Response 1:

We thank you for the time and energy you spent reviewing our manuscript. We also thank you for this suggestion. We agree with you and have developed the English by being reviewed by an experienced editor.

Point 2: In the studies, Synaptic acid is given together with Scopolamine. Grape Juice and SA has been shown to have an effect on CYP enzymes. Maybe the effect on memory os not due to inflammation or other effects but because of the inhibition of CYP enzymes. Scopolamine is interacting with CYP3A4 one of the targets of SA. The authors should test if the effect of SA is not due to the effect on the degradation or activation of Scopolamine.

Response 2:

Thank you for providing these insights. The liver and small intestine have the highest CYP3A4 activity. Some important CYP3A4 interactions are due to intestinal rather than hepatic enzyme inhibition (eg, grapefruit). CYP3A4 was also expressed by blood–brain barrier (BBB) endothelial cells and by the majority of neurons (75 ± 10%) (https://doi.org/10.1111/j.1528-1167.2010.02956.x). It has also been increasingly recognized that inflammatory mediators associated with a range of disease states are capable of having profound effects on CYP3A4 gene expression. Patients with inflammation, particularly elevated acute phase proteins such as C-reactive protein (CRP) have been noted to have reduced CYP3A4 function (https://doi.org/10.3390/pr10061066). We agree with your points, and we think this is an interesting point to be addressed, but we haven’t conducted this study. In further study, we would like to find out whether the degradation or activation of scopolamine is affected by measuring the expression of CYP3A4.

Point 3: One other test that should be added is to test if SA has the same effect when not given at the same time, but where first scopolamine is inducing an effect on memory and than SA is given. to show that the effect of SA is on the biological processes like inflammation and COX.

Response 3:

Scopolamine does not affect in the long term. Because it affects in the short term, even if scopolamine is given first, the effect does not last long. Therefore, it is not certain that the effect of scopolamine will continue until sa is processed later. In addition, we focused on the effect of sinapic acid on the inhibition and prevention of scopolamine. however, we have the plan to conduct the test when scopolamine and sinapic acid are not given at the same time in consideration of your opinion in future research.

Reviewer 2 Report

The manuscript "Effect of Sinapic acid on scopolamine-induced learning and memory impairment in SD Rats" is a search for more effective therapy in AD. The topic is justified but distant from the clinic.

My comments:

Abstract-

Alzheimer's disease accounts for  about 50% of senile dementia.

Acetylcholine esterase inhibitor can be applied to the treatment rather of PDD

Introduction –

What genetic factors are the authors referring to? Whether FAD or SAD.

…several mechanisms are involved in AD, including oxidative stress, cholinergic deficit, neuroinflammatory, cell death, and genetic factors. Are all these factors equivalent?

Materials-

5-weeks-old male rats are not young adult rats. 3.0-3.5 month-old rats are young adult animals. The experiment investigated the developmental changes in which programmed cell death, PCD, occurs.

These are not age changes.

How many groups of animals were in the experiment?

From how many animals were Hippocampus extracts prepared?

Discussion

The discussion is a retelling of the results and should be edited.

Summarize the results in detail in the conclusion.

Correct language errors.

Author Response

Response to Reviewer 2 Comments

We thank you for your time and consideration of our manuscript. Below we address your comments and list of changes we made to our manuscript according to your reports. The original comments are provided in blue color, whereas our answers are given in black. The appropriate changes made in the revised manuscript are highlighted.

We have taken the opportunity to make some other changes in order to make corrections and refine the quality of the manuscript.

Point 1:

Abstract-

Alzheimer's disease accounts for about 50% of senile dementia.

Acetylcholine esterase inhibitor can be applied to the treatment rather of PDD

Response 1:

We thank you for the time and energy you spent reviewing our manuscript. According to Centers for Disease Control and Prevention of U.S. Department of Health & Human Services, Alzheimer’s disease is the most common cause of dementia, accounting for 60 to 80 percent of cases. Patients with Parkinson's disease dementia (PDD) often have significant cholinergic defects, which may be treated with cholinesterase inhibitors (ChEIs). These deficits are largely localized to the cholinergic system of the basal forebrain and brainstem, in contrast to patients with Alzheimer's disease (AD), where cholinergic deficits are primarily seen in the hippocampus. In this manuscript, we focused on the cholinergic deficits in the hippocampus.

Point 2:

Introduction-

What genetic factors are the authors referring to? Whether FAD or SAD.

…several mechanisms are involved in AD, including oxidative stress, cholinergic deficit, neuroinflammatory, cell death, and genetic factors. Are all these factors equivalent?

Response 2:

Both sporadic and familial AD forms share main neuropathological characteristics, namely β-amyloid plaques, neurofibrillary tangles, and regionalized neuronal loss. However, The neuropathology seen in familial AD does not differ in form from sporadic disease, but is generally more severe, especially in regard to the deposition of Aβ.

And we revised the phrase you pointed out to be a clearer expression. “~recent studies have shown that several mechanisms are involved in the deposition of β- amyloid (Aβ) plaques, Aβ aggregation, neuronal death, cholinergic nervous system diseases, the formation of NFTs with abnormal phosphorylated tau, oxidative stress, neuroinflammation, and genetic factors”.

Point 3:

Materials-

5-weeks-old male rats are not young adult rats. 3.0-3.5 month-old rats are young adult animals. The experiment investigated the developmental changes in which programmed cell death, PCD, occurs.

These are not age changes.

How many groups of animals were in the experiment?

From how many animals were Hippocampus extracts prepared?

Response 3:

We purchased 5-week-old rats, adapted them for 1 week, administered sinapic acid for 10 days, and finally conducted behavioral experiments at 8-week-old of age. Because the rats were induced memory loss by injecting scopolamine and we investigated the preventive effect of sinapic acid, the age of the rat was not greatly considered and the experiment was performed by adjusting the appropriate weight of the rat through other references.

We experimented with 4 groups (control, scop, sa, scop+sa) of 6 animals in each group. Hippocampus extracts were also prepared from 6 rats.

Point 4:

Discussion-

The discussion is a retelling of the results and should be edited.

Summarize the results in detail in the conclusion.

Response 4:

We agree with you and have reflected on this comment by rewriting the discussion section. We have added the summary of the results in detail in the conclusion.

Round 2

Reviewer 1 Report

In the figures specifically figure legend 7: **p < 0.01 vs. the control group. 

This can not be that the control is significantly different from the control 

and figure legend 8: **p < 0.01 vs. the control group. ##p < 0.01 vs. the SCOP group. 365

Analyzed by one-way ANOVA with LSD test.

This is a copy paste mistake from figure 7. Please correct this.

The English has improved. I am ok with the answer on point 2, but I do not agree with the answer to point 3. In the slices study it takes 1 hour before the effect of scopolamine is visible, and this effect takes all the way till the end of the study. Here the authors could test if we give scopolamine first and then after 30 min, 45 min or 1 hour SA, do we see the effect of SA or do you need SA present before or at the same time as Scopolamine? In addition, if we give SA first for 30, 45 or 1 hour before we give scopolamine (to compare better with the rat studies where SA was given 10 days before ) is there an additive effect.

Author Response

Thank you for the mistaken description. We checked the errors in Figure 7. The p-value display of the control was moved to the SCOP group as compared between the control and SCOP.

We understood the content of your proposed experiments. Like your suggestion, the way that SCOP administration first and then SA administration can see the recovery of SA in the decrease caused by SCOP. Moreover, for the same environment as the behavioral test, SA should be processed first, and then SCOP should be processed. However, in the LTP experiment, the valid fEPSP activity is measured 30min after the high-frequency stimulation. Even if SCOP and SA are separately processed, they are eventually processed before the high-frequency stimulation. Therefore, whether the results differ from simultaneously processing SCOP and SA is uncertain. Because of this limitation on the protocol of LTP, many studies using organotypic have chosen the method of simultaneous processing (Kim et al., 2022; Park et al., 2021). We think measuring spontaneous activity is more appropriate to your suggestion than LTP. In addition, to proceed with the experiment under the suggested conditions, the process with incubation of SCOP treatment, LTP measurement, incubation of SA treatment again, and LTP measurement is more exact than inputting drugs with aCSF while recording. It is expected that it will be an accurate experiment to proceed (Ahuja et al., 2007). If these proceeds, the protocols of the control, SA, and SCOP-only groups will also need to be modified. Therefore, if given the opportunity, we would like to design an experiment on the effect of SA by conducting an experiment by SCOP first, SA post/ SA first, SCOP post/ 30,45,60 min hour in the follow-up study.

Kim, K. J., Hwang, E. S., Kim, M. J., Rha, C. S., Song, M. C., Maeng, S., Park, J. H., & Kim, D. O. (2022). Effects of Phenolic-Rich Pinus densiflora Extract on Learning, Memory, and Hippocampal Long-Term Potentiation in Scopolamine-Induced Amnesic Rats. Antioxidants (Basel, Switzerland)11(12), 2497. https://doi.org/10.3390/antiox11122497

Park, H. S., Hwang, E. S., Choi, G. Y., Kim, H. B., Park, K. S., Sul, J. Y., Hwang, Y., Choi, G. W., Kim, B. I., Park, H., Maeng, S., & Park, J. H. (2021). Sulforaphane enhances long-term potentiation and ameliorate scopolamine-induced memory impairment. Physiology & behavior238, 113467. https://doi.org/10.1016/j.physbeh.2021.113467

Ahuja, T.K., Mielke, J.G., Comas, T., Chakravarthy, B. and Mealing, G.A.R. (2007), Hippocampal slice cultures integrated with multi-electrode arrays: a model for study of long-term drug effects on synaptic activity. Drug Dev. Res., 68: 84-93. https://doi.org/10.1002/ddr.20170
